# The Impact of Sample Quantity, Traceability Scale, and Shelf Life on the Determination of the Near-Infrared Origin Traceability of Mung Beans

**DOI:** 10.3390/foods13203234

**Published:** 2024-10-11

**Authors:** Ming-Ming Chen, Yan Song, Yan-Long Li, Xin-Yue Sun, Feng Zuo, Li-Li Qian

**Affiliations:** 1College of Food Science, Heilongjiang Bayi Agricultural University, Daqing 163319, China; chenmingming515@163.com (M.-M.C.); seany_lib@163.com (Y.S.); a99008191@163.com (Y.-L.L.); 15245806261@163.com (X.-Y.S.); zuofeng-518@126.com (F.Z.); 2National Coarse Cereals Engineering Research Center, Daqing 163319, China; 3Key Laboratory of Agri-Products Processing and Quality Safety of Heilongjiang Province, Daqing 163319, China

**Keywords:** mung bean, near-infrared spectroscopy, traceability model, influential factors, discriminatory results

## Abstract

This study aims to address the gap in understanding of the impact of the sample quantity, traceability range, and shelf life on the accuracy of mung bean origin traceability models based on near-infrared spectroscopy. Mung beans from Baicheng City, Jilin Province, Dorbod Mongol Autonomous, Tailai County, Heilongjiang Province, and Sishui County, Shandong Province, China, were used. Through near-infrared spectral acquisition (12,000–4000 cm^−1^) and preprocessing (Standardization, Savitzky–Golay, Standard Normal Variate, and Multiplicative Scatter Correction) of the mung bean samples, the total cumulative variance contribution rate of the first three principal components was determined to be 98.16% by using principal component analysis, and the overall discriminatory correctness of its four origins combined with the K-nearest neighbor method was 98.67%. We further investigated how varying sample quantities, traceability ranges, and shelf lives influenced the discrimination accuracy. Our results indicated a 4% increase in the overall correct discrimination rate. Specifically, larger traceability ranges (Tailai-Sishui) improved the accuracy by over 2%, and multiple shelf lives (90–180–270–360 d) enhanced the accuracy by 7.85%. These findings underscore the critical role of sample quantity and diversity in traceability studies, suggesting that broader traceability ranges and comprehensive sample collections across different shelf lives can significantly improve the accuracy of origin discrimination models.

## 1. Introduction

The implementation of traceability technology to determine the origin of agricultural products can provide a technical and theoretical basis for the tracing and confirmation of geographical indication products, as well as those of a regional or specialty nature that has gained a reputation. The methodologies and conceptual frameworks underpinning this technical research were initially developed, and a technical system was gradually being established. The feasibility of near-infrared (NIR) spectroscopy has been widely studied in the context of traceability technology for the origin of agricultural products, both domestically and abroad. For example, Son et al. explored its use in rice [1]. Sweet potatoes were studied in 2022 [2], followed by apples [3], edible oils [4], camellia oils [5], mung beans [6], and coffee [7]. Qian et al. used near-infrared spectroscopy to collect spectral data from five rice origins, namely Wuchang and Jiamusi in Heilongjiang Province, etc. The data were then subjected to partial least squares (PLS) analysis to establish an origin discrimination model. The results demonstrated that the accuracies in discriminating the origin of rice were 95.83%, 100.00%, and 95.83%, respectively, with an overall discriminating accuracy exceeding 95.00%. This study on the origin traceability of rice was conducted using the NIR spectral analysis technology [8]. In a study by Liu et al., near-infrared spectroscopy was used to collect spectral data from red Fuji apples sourced from three geographical locations (Xinjiang, Shanxi, and Shandong). Before analysis, these data underwent preprocessing steps, including normalization and multivariate scattering correction (MSC), etc. A model for origin discrimination was developed using principal component analysis (PCA) with the K-nearest neighbor (KNN) method. The results demonstrated that the accuracies of discriminating the origin of red Fuji apples reached 97.30% and 92.30%, respectively, effectively achieving origin traceability of red Fuji apples [9]. Xia et al. assessed shiitake from disparate origins using near-infrared spectroscopy. They then established discriminant models for the provinces of Jilin, Hubei, and Fujian by preprocessing the raw spectra and combining them with the PLS discriminant analysis method. The results demonstrated that the discrimination accuracies of the model were 96.70%, 95.60%, and 100.00% for the Jilin, Hubei, and Fujian Provinces, respectively. The establishment of this method provides a novel approach for authenticating the origin of shiitakes [10].

In a discriminative study of palm sugar using near-infrared spectroscopy, Rismiwandira et al. developed a discriminative model for the adulteration of palm sugar by combining PCA and PLS regression analysis after five preprocessing methods (First Derivatives, Vector Normalization, Standard Normal Variable (SNV), MSC, and Baseline). The results demonstrated that the accuracy of discriminating palm sugar adulteration was 91.00% [11]. Zhuang et al. collected raw spectral data from 60 green tea samples using near-infrared spectroscopy to improve the performance of the discriminant model through Smoothing, Derivatives, Vector Normalization (VN), and PCA. A discriminative model was constructed using a Support Vector Machine (SVM). The results demonstrated that the discrimination accuracy of green tea samples was 96.11%. The combination of near-infrared spectroscopy with appropriate chemometrics is an effective method for determining the origin of green tea [12]. NIRS utilizes specific functional groups (such as C–H, O–H, N–H) in substances to selectively absorb near-infrared light. Different geographical locations and growth conditions can affect the distribution pattern of these functional groups in substances, forming unique spectral features. By analyzing these spectra, the specific chemical compositions of different regions can be reflected, forming the theoretical basis for near-infrared spectroscopy for origin tracing. Thus, it is possible to distinguish the geographical origins of different research objects and purposes using NIR spectroscopy. However, this technology is affected by numerous factors. (a) Equipment quality: If the light source of NIR equipment is unstable, the intensity and wavelength of the emitted near-infrared light may fluctuate, making it difficult to accurately measure the absorption and reflection characteristics of the sample to light. (b) Calibration issues: When analyzing a certain chemical substance, wavelength calibration errors can cause the position of characteristic absorption peaks to shift, and the model may not be able to correctly identify them. (c) Environmental interference (e.g., humidity, temperature): When the humidity is high, the surface of optical components (lenses, mirrors, etc.) in NIR devices may adsorb water vapor, causing the focusing position of light passing through the lens to shift, resulting in changes in the interaction between light and the sample and thereby affecting the collection of spectral data. When the temperature rises, it may cause inaccurate focusing of light, affecting the accuracy of spectral measurements and ultimately reducing the accuracy of the model. Meanwhile, the application prospects of NIR spectral analysis models in the field of agricultural product origin tracing are broad, but they also face challenges such as overfitting, model robustness, and potential variability under different conditions: (a) Overfitting is one of the common problems in machine learning. In NIR spectral analysis, overfitting may be caused by various factors (insufficient training data, excessive or underfitting model parameters, etc.). (b) The model robustness may be affected by various factors (sensitivity to differences in sample type, composition, and structure), which may lead to a decrease in the predictive accuracy of the model. (c) There is potential variability under different conditions (growing seasons or environmental factors), as the growth rate and quality of the sample may change, which will affect the accuracy of the NIR spectral analysis model. The model needs to be able to adapt to these changes in order to maintain high prediction accuracy under different conditions.

In a study by Qian et al., NIR diffused reflectance spectroscopy was used to investigate the origin traceability of rice samples sourced from the Jiansanjiang and Wuchang regions of China. The results demonstrated 100.00% and 98.00% accuracies in discriminating the origin traceability model based on Jiansanjiang and Wuchang rice, respectively [13]. Similarly, Firmani et al. developed a discriminant model for black tea of different origins based on NIR spectra, achieving an accuracy of 98.57% through the use of NIR spectroscopy to differentiate between black tea from Dajiling and other regions using Partial Least Squares Discriminant Analysis (PLS-DA). The findings of this study indicate a robust correlation between the accuracy of origin discrimination models and geographical origins [14]. Zhang et al. used near-infrared spectroscopy to distinguish six tea categories: oolong, black, white, green, black, and yellow. The results indicated that, except for yellow tea, the spectral characteristics of the other tea types exhibited considerable variation in the three-dimensional spatial distribution. This variation may be attributed to differences in processing techniques and tea tree varieties [15]. Yu et al. used near-infrared spectroscopy to gather spectral data from Citri reticulata pericarpium samples at the ages of 5, 10, 15, 20, and 25 years. Following the application of various spectral preprocessing techniques with Fisher ’s linear discriminant (FLD) analysis, a discriminant model was constructed to differentiate between the samples of Citri reticulata pericarpium from different years. The discrimination accuracies for the Citri reticulata pericarpium samples were 90.00%, 100.00%, 90.00%, 100.00%, and 100.00%, respectively, indicating that different years affected the results of the traceability model based on near-infrared spectroscopy [16]. Previous studies have demonstrated that different types of agricultural products analyzed using near-infrared spectroscopy exhibit variations in chemical composition and physical properties, which can be attributed to various factors, including their origin, variety, year, and processing technology. An appropriate model was established using the characteristic information obtained by near-infrared spectroscopy, and the impact of these factors on the results of the discernment of the origin was analyzed. However, there is a paucity of research on the effects of the distance range of the sample origin, sample quantity collected, and shelf life of the samples on the discrimination results of the origin traceability. Consequently, it can be regarded as a pivotal aspect of the investigation into the influence of traceability technology on agricultural products, and its variables can be evaluated for their impact on the original traceability model.

In this study, we analyzed the results of near-infrared spectroscopy (NIRS) to discriminate between mung bean origins. Samples of mung beans were collected from four sources: the Dorbod Mongol Autonomous, Tailai County of Heilongjiang Province, Baicheng City of Jilin Province, and Sishui County of Shandong Province. The raw spectra of the mung bean samples were subjected to spectral preprocessing using a range of methods. A near-infrared (NIR) discrimination model for mung bean samples of different origins was established using PCA combined with the KNN method to confirm the feasibility of tracing mung bean samples based on NIR spectroscopy. Furthermore, the impact of the sample quantity, traceability range, and shelf life on the establishment of different mung bean origin traceability discriminant models was analyzed. An analysis of the influence of the aforementioned factors on the mung bean origin discrimination model offers a theoretical foundation for the establishment of a technical standard system for the traceability of diverse agricultural products, including the traceability of high-value-added agricultural products.

## 2. Materials and Methods

### 2.1. Sample Collection

Mung bean samples were collected one week before the 2021 harvest using a representative checkerboard sampling method specifically tailored to diverse origins and agricultural range settings. The sampling locations were established at five strategic points—east, west, south, north, and center—within each region [17]. Test samples were randomly collected from designated points in Tailai County (Tailai) and Dorbod Mongol Autonomous County (Dorbod Mongol Autonomous) in Heilongjiang Province, Baicheng City (Baicheng) in Jilin Province, and Sishui County (Sishui) in Shandong Province. Each region is associated with a distinct geographical indication of mung beans. Approximately 2.0–3.0 kg of pods was collected from each site. A GPS device (NEO-6M-0-001, Shenzhen Pengrunfa Electronics Co., Shenzhen, China) was used to record detailed information, including collection sites, varieties, and geographic coordinates (longitude and latitude) (Table 1).

### 2.2. Sample Pretreatment

The mung bean samples were placed in a dust-free and clean environment for uniform drying and the removal of husks, minute dust particles, and debris. This was performed to ensure that the beans were intact. The samples were subjected to a grinding process using a cyclone mill (CT193Cyclotec^TM^, Guangzhou Easy Measurement Instrument Co., Ltd., Guangzhou, China) and then sieved through a 100-mesh filter to ensure homogeneity. All mung bean samples were placed in sealed bags and stored in a constant-temperature and -humidity box (temperature 25 °C, relative humidity 45%) (ZXDP-B2080, Shanghai Zhicheng Analytical Instrument Manufacturing Co., Ltd., Shanghai, China) until the end of the experiment.

### 2.3. Spectra Acquisition

NIR spectra were obtained using a Fourier-transform NIR spectrometer (TENSOR, Bruker Beijing Technology Co., Ltd., Beijing, China) [18]. Before commencing the measurements, the spectrometer was turned on, and the samples were pre-warmed for one hour. The sensor could operate over a scanning range of 12,000–4000 cm^−1^, with a scan frequency of 64 and a resolution of 8 cm^−1^. For the collection of spectral data, 100 g of powdered mung beans was placed in a revolving sampling cup. The measurements were conducted at an ambient temperature of approximately 25 °C and relative humidity of approximately 45%. To prevent cross-contamination between the samples, the sampling cup was cleaned after each measurement. The background single-channel spectra, calibration signals, and peak positions were meticulously recorded to eliminate noise interference from instrument operation. Each NIR sample was scanned 15 times, and the mean of the three spectra was used to construct the final spectrum, resulting in 600 spectral data points.

### 2.4. Spectra Preprocessing

Several techniques were used in the NIR spectral preprocessing stage to enhance the precision and predictive capacity of the origin discrimination algorithms. In accordance with ref. [6], these included Standardization (SS), Savitzky–Golay (SG) [19], Standard Normal Variate (SNV) [20], and Multiplicative Scatter Correction (MSC) [21], among others. Standardization centralizes the mean value of the original spectrum and eliminates the influence of deviations caused by dimensional differences, minor self-variations, and significant numerical disparities. Savitzky–Golay of the original spectrum aids in reducing noise, enhancing the uniformity of the spectrum, and thereby mitigating the impact of noise. SNV is primarily employed to diminish interference arising from solid particle mass, surface scattering, and variations in the optical path during the measurement of spectral signal data. MSC is designed to eliminate the effects of scattering phenomena caused by uneven particle sizes or differences in sample particle sizes in the spectral data. Consequently, this study primarily selected these four methods for the spectral preprocessing of near-infrared (NIR) original spectra.

### 2.5. Modeling of Influencing Factors

#### 2.5.1. Sample Quantity

PCA was conducted using near-infrared spectra derived from small (*n* = 200), medium (*n* = 400), and large (*n* = 600) mung bean samples from disparate origins, designated A, B, and C, respectively. The optimal spectral preprocessing method (Standardization) was applied to the original spectral data for different sample quantities (*n* = 200, 400, and 600). A near-infrared discrimination model was established by combining it with the KNN method to analyze the influence of sample quantity on the near-infrared traceability discrimination model of mung bean origin. This model was developed using various sample quantities (*n* = 200, 400, and 600).

#### 2.5.2. Traceability Scale

Taking the origin of Tailai mung beans as the reference point, the near-infrared spectra based on mung bean samples of different origins were used to perform principal component analysis on the traceability scales of larger (Tailai-Sishui) and smaller (Tailai-Baicheng, Tailai-Dorbod Mongol Autonomous) origins. These origins were designated as A and B, respectively. In addition, the spectral preprocessing method of Standardization was used, and the KNN method was integrated to construct two NIR discriminant models, one encompassing the larger (Tailai-Sishui) and smaller (Tailai-Baicheng and Tailai-Dorbod Mongol Autonomous) origin traceability scales, and the other analyzing the impact of sample traceability scales on the discriminant NIR mung bean origin traceability model.

#### 2.5.3. Shelf Life

Mung bean samples from four origins, namely Tailai, Baicheng, Dorbod Mongol Autonomous, and Sishui, were selected for analysis. The samples were scanned at 90-day intervals for four measurements: 90, 180, 270, and 360 d. The mung bean samples of different origins were placed in a cool and dry environment, with temperatures ranging from 15 °C to 20 °C and relative humidity ranging from 50% to 60%. NIR discriminant models for the shelf life of mung beans of different origins (90, 180, 270, 360, and 90–180–270–360 d) were established by combining the KNN method and labeling them into five categories, namely, T_1_, T_2_, T_3_, T_4_, and T_1_–T_2_–T_3_–T_4_. To assess the accuracy of the model in differentiating between the origins of 20 randomly selected mung bean samples collected from outside the four origin modeling scenarios with varying shelf lives, the impact of shelf life on the efficacy of the near-infrared origin traceability discrimination of mung beans was evaluated.

### 2.6. Statistical Analysis

Raw spectral data from the NIR techniques were organized using Microsoft Excel 2010 and Origin 2019b. The samples were divided into training and validation sets at a ratio of three to one to ensure rigorous testing and validation of the models. Dimensionality reduction of the near-infrared spectral data using PCA combined with the KNN method was carried out to establish the near-infrared discriminant model (Python). PCA is used to transform a set of correlated variables into a set of uncorrelated principal components (PCs) that explain the greatest possible amount of variation in the data, as well as to provide comprehensive data visualization [22]. The most commonly used software for principal component analysis is IBM SPSS Statistics 26.0, Origin 2021 Pro, Matlab R2024b, Python 3.11.6, etc. In Python, the PCA class in the Scikit-learn can be used to implement its analysis. Upon applying PCA to the analytical data for four geographical origins, the mung bean samples could be preliminarily clustered (the first three PCs), and the data used in the analysis were normalized.

## 3. Results and Discussion

### 3.1. Near-Infrared Spectroscopic Origin Traceability Study of Mung Beans of Different Origins

#### 3.1.1. NIR Raw Spectral Acquisition and Preprocessing

To effectively distinguish the mung bean samples from four different origins, namely Tailai, Baicheng, Sishui, and Dorbod Mongol Autonomous, the collected mung bean samples of four origins were subjected to near-infrared (NIR) spectroscopy. The original NIR spectrograms of the mung beans of different origins are shown in Figure 1a. However, in the context of near-infrared (NIR) spectral analysis, the raw spectra of complex and irregular spectra must be preprocessed to eliminate the influence of certain factors. As illustrated in Figure 1b, the preprocessed NIR spectra of the mung beans exhibited a reduction in baseline drift and signal fluctuation compared to the original NIR spectra of mung beans. This improvement was achieved by applying four spectral preprocessing methods: SS, SG, SNV, and MSC. Standardization places all variables on the same order of magnitude, avoiding erroneous conclusions due to different measurement units between variables and improving accuracy; SG reduces the impact of noise on principal component extraction, while MSC eliminates the influence of scattering effects on principal component extraction, making the principal components more representative of the main features of the data and extracting clearer and more stable principal components, thereby improving the accuracy of the subsequent analysis; SNV improves the quality of data, making the main components more reflective of the true structure of the data, and can enhance the accuracy of PCA. In summary, different spectral preprocessing methods have varying impacts on the results of PCA. However, these improvements are difficult to discern without the aid of a visual representation, except for the standardization of near-infrared spectral data of mung beans. Consequently, mung bean NIR spectral data following different spectral preprocessing steps must be subjected to PCA to construct subsequent model inputs. This allows for the comparison of the predictive performance of models based on different input variables and the evaluation of the preprocessing performance of this spectral preprocessing method for the mung bean NIR spectra.

#### 3.1.2. PCA of Near-Infrared Spectra of Mung Beans of Different Origins

A comparative analysis of the mung bean samples of different origins, preprocessed with near-infrared (NIR) spectra, was conducted using PCA. The NIR spectral data were subjected to preprocessing using the four processing methods. To determine the optimal number of principal components, the principal component variables were screened, and we drew the principal component rubble map (Figure 2), from which the first three principal components can be extracted.

The cumulative variance contributions of the various spectral preprocessing methods to each principal component of the NIR spectra are presented in Table 2. As illustrated in Table 2, the Standardization method exhibited the most substantial total cumulative contribution among all the NIR spectral preprocessing methods. The first, second, and third principal component contributed 81.56%, 12.82%, 3.78%, and the total cumulative variance contribution of the three principal components reached 98.16%. Accordingly, the mung bean near-infrared spectral data processed using the Standardization method were selected as input variables to establish an origin discrimination model in this study.

#### 3.1.3. KNN Analysis of Near-Infrared Spectra of Mung Beans of Different Origins

A near-infrared discrimination model of the mung bean samples of different origins was established based on the PCA feature extraction information and standardized spectral preprocessing of the samples. This was achieved by combining them with the KNN method. The discrimination accuracy of the model was evaluated using a K-fold cross-validation approach (k = 4), wherein 600 spectral datasets were obtained and divided into four subsets. One subset was designated as the validation set to assess the model’s discrimination accuracy, whereas the remaining three subsets (k-1) were used as the training set (sample set) to ascertain the model’s precision and reliability. As illustrated in Table 3, to further examine the discriminative accuracy of the origin traceability, the feature information extracted by PCA with the KNN method was used to construct an NIR-discriminative model for mung beans of diverse origins. The accuracy of the mung bean origin determination for Dorbod Mongol Autonomous and Tailai was 100.00%; the accuracies of mung bean origin determination for Baicheng and Sishui were 97.22% and 97.50%, respectively, resulting in an overall origin determination accuracy of 98.67%.

Ren et al. collected and preprocessed the original spectra of different Keemun black teas using NIRS combined with an SVM to establish a discriminant model of different Keemun black teas, with an accuracy of 99.01% [23]. Similarly, Li et al. used NIRS with PCA and Fisher’s linear discrimination analysis (FLDA) to construct a discrimination model for different millet origins. The overall discrimination accuracy for different millets was 92.30% [24]. Tian et al. used NIRS with an Artificial Neural Network (ANN) method to develop a soybean origin discrimination model for various countries. The overall discrimination accuracy for soybeans from disparate countries was 95.65%. This indicates that NIRS, when used with chemometric analytical techniques, can be used to distinguish the provenance of diverse types of agricultural commodities [25]. This study demonstrates that it is feasible to discriminate mung beans of different origins using NIRS with the KNN method.

The performance index was calculated by predicting data from the test set using the trained model to evaluate the discriminative model built with NIR spectral data. Common model evaluation measures include the F_1_-score, precision, recall, and accuracy. Precision is defined as the percentage of true positive predictions among all positive predictions. Recall is the proportion of true positive predictions out of all positive cases. Higher values of precision and recall indicate a better discriminative ability of the model between positive and negative samples. Accuracy, a metric for evaluating classification problems, generally refers to the overall correctness of the model. The closer the logarithmic value is to 1, the higher the model’s recall or accuracy is. The F_1_-score, which harmonizes precision and recall (2/F_1_ = 1/P + 1/R), is used because precision and recall often present trade-offs; high precision usually corresponds with low recall and vice versa. Thus, the F_1_-score serves as a criterion for assessing the comprehensive performance of the model. Table 4 details the assessment indices for various mung bean NIR-discriminating models. The table indicates that the accuracy of the model for different origins of mung bean samples was 98.25%, closely aligning with the correct rate of origin discrimination, with all F_1_-scores exceeding 0.97.

### 3.2. Effect of Sample Quantity on the Accuracy of Near-Infrared Mung Bean Origin Traceability Discrimination

Researchers have collected varying quantities of samples to develop an original traceability model for agricultural products. The quantity of samples used in the modeling ranged from a dozen to several hundred, and the resulting discriminative effects varied. A larger sample (600) of products from different origins were collected in the previous period for a discriminative analysis of their origin traceability. The objective of this study was to ascertain whether a smaller sample would have a significant impact on the determination of origin traceability using NIR spectroscopy. Based on refs. [26,27], most studies used a proportional increase (1:2:3) in sample quantity to analyze the impact of different origin traceability discrimination results. Therefore, a smaller, medium, or larger amount of samples were selected (randomly) from the collected near-infrared spectral data of mung beans of different origins to reduce errors. For this purpose, the spectral data of the samples belonging to classes A (*n* = 200), B (*n* = 400), and C (*n* = 600) were used to compare the discriminative analysis.

#### 3.2.1. Effect of Sample Quantities on Origin Discrimination Results Based on PCA

The cumulative variance contributions of the different sample quantities (A, B, and C) to each principal component of the mung bean NIR spectra are presented in Table 5. The distributions of the mung bean feature vectors extracted by spectral preprocessing for the principal components (PCs) of the different sample quantities are shown in Figure 3. As illustrated in Table 5 and Figure 3, the cumulative variance contributions of the first and second PCs derived from the sample spectral data of classes A (*n* = 200), B (*n* = 400), and C (*n* = 600) were 84.42%, 90.39%, and 94.38%, respectively. The cumulative variance contribution of the third principal component was 3.77%–3.78%. The total cumulative variance contributions of the three PCs reached 88.19%, 94.17%, and 98.16% (C > B > A), respectively. This indicated that discrepancies between the various sample quantities were primarily observed in the first and second PCs. Consequently, to further investigate the impact of varying sample quantities on the outcomes of mung bean origin classification, it is essential to conduct a KNN analysis on the PCA results of mung bean spectral data from diverse origins and use the same KNN method to develop prediction models for the origin classification of varying sample quantities.

#### 3.2.2. Effect of Sample Quantities on Origin Discrimination Results Based on KNN Analysis

The PCA feature extraction information from the spectral preprocessing of mung bean samples with varying quantities (*n* = 200, 400, and 600) was used to construct NIR discriminant models of mung bean samples in different quantities. This was achieved by integrating the KNN method. The same 3.1.3 method was used to generate the results shown in Table 6, which presents the classification results for different sample quantities of mung beans. Figure 4 shows the corresponding contour maps. As illustrated in Table 6 and Figure 4, the overall discrimination accuracies of the origin discrimination were 94.00%, 97.00%, and 98.67% for classes A (*n* = 200), B (*n* = 400), and C (*n* = 600), respectively. Among the results of mung bean origin discrimination obtained using 200 samples, three samples were misidentified. One sample from Tailai was misclassified as Sishui, one sample from Baicheng was misclassified as Sishui, and one sample from Sishui was misclassified as Baicheng. The results of the mung bean origin discrimination obtained using 400 samples revealed two misclassified samples: one from Baicheng misclassified as Sishui, and one from Sishui misclassified as Baicheng. Similarly, the results of the mung bean origin discrimination obtained using 600 samples revealed two misclassified samples: one from Baicheng misclassified as Sishui, and one from Sishui misclassified as Baicheng. A comparison of the results of the mung bean origin discrimination with different sample quantities revealed that the correct rate of mung bean origin discrimination with multiple sample quantities (*n* = 600) was higher than with fewer sample quantities (*n* = 200).

In their study, Qiu et al. quantified the mineral element content in 40, 80, and 120 mung bean samples using ICP-MS and established the discrimination accuracies of the discrimination of mung bean samples of different origins using principal component and PLS discriminant analyses based on the models established using different quantities of samples. The results showed that the accuracy of the mineral element discrimination model established using a larger sample quantity (*n* = 120) for different mung bean origins was 96.70%, while the accuracy of the discrimination model with a smaller sample quantity (*n* = 40) was higher than 1.70% [26]. This indicates that the higher the sample size is, the higher the accuracy of the technical prediction is. This article is consistent with the above research results. The accuracy of the near-infrared discrimination model for different mung bean origins established using a larger sample quantity (*n* = 600) is 98.67%, and the accuracy of the discrimination model with a smaller sample quantity (*n* = 200) is higher than 1.97%. The results of the origin discrimination were affected by the quantity of samples collected. As the quantity of samples increased, the information obtained became more representative of the origin.

To assess the efficacy of the discriminant models constructed from NIR spectral data with varying sample quantities, the evaluation indices of the NIR discriminant models for mung beans with distinct sample quantities are presented in Table 7. As illustrated in Table 7, the accuracies of the models developed using mung bean samples with sample quantities A (*n* = 200), B (*n* = 400), and C (*n* = 600) were 93.00%, 96.75%, and 98.25%, respectively. The model developed using the sample quantity C exhibited the highest accuracy (C > B > A). The F_1_-scores of the mung bean origin discrimination model for A (*n* = 200) were all above 0.90, whereas the F_1_-scores of the mung bean origin discrimination model for B (*n* = 400) were all above 0.92. The F_1_-scores of the mung bean origin discrimination model for C (*n* = 600) were all above 0.97. A comparison of the evaluation indices of the mung bean origin discrimination model with different sample quantities indicated that the discrimination accuracy of the model established with more origin samples was higher than the model established with fewer samples. This suggests that an increase in the sample quantity correlates with enhanced accuracy in the discriminatory model of the original samples. Accordingly, the choice of sample quantity represents a pivotal aspect in the investigation of traceability methodologies pertaining to the provenance of agricultural commodities.

### 3.3. Effect of Sample Traceability Scale on the Accuracy of Mung Bean Near-Infrared Origin Traceability Determination

Researchers and scholars have studied the impact of analyzing the traceability range of samples collected from diverse sources using IR-MS and ICP-MS techniques to distinguish their origins. For instance, He et al. distinguished between Maca (δ^13^C, δ^15^N, δ^2^H, and δ^18^O) on large (Peru and China) and small (Yunnan, Xinjiang, and Tibet) scales by using stable isotope techniques [28]. Qiu et al. distinguished mung beans (B, Na, Mg, P, K, Ca, etc.) from a large scale (Northeast and Shandong) and from a small scale (Northeast: Tailai, Dorbod Mongol Autonomous, and Baicheng) using the mineral element technique. However, the impact on the scale of traceability of samples based on the NIR spectroscopic technique has not been reported [26]. Therefore, the effects of different sample traceability scales on the origin discrimination of mung beans were investigated using NIR spectroscopy. The spectral data from the mung bean samples of class A (Tailai-Sishui) and class B (Tailai-Baicheng and Tailai-Dorbod Mongol Autonomous) origins were used for discriminant analysis. The actual distances from the origins of the various categories of mung bean samples are shown in Figure 5.

#### 3.3.1. Effect of Traceability Scales on Origin Discrimination Results Based on PCA

##### Impact of Larger Scales on the Results of Origin Determination

As illustrated in Figure 6, the sum of the contribution rates of the first and second PCs in A (Tailai-Sishui) exceeded 90.00%. The cumulative variance contribution rate of the first principal component was 66.84%, whereas that of the second principal component was 27.69%. This indicates that the larger traceability scale differentiated the two origin mung bean samples effectively, and that the first two extracted PCs could be used in the discriminant analysis of the two origins of Tailai-Sishui.

##### Impact of Smaller Scales on the Results of Origin Determination

As illustrated in Figure 7, the sum of the contribution rates of the first and second PCs in B (Tailai-Baicheng) exceeded 75.00%. The cumulative variance contribution rate of the first principal component was 54.26%, whereas that of the second principal component was 21.12%. One of the Tailai samples was distinguished from the Baicheng sample (Figure 7a). The sum of the contribution rates of the first and second PCs in B (Tailai-Dorbod Mongol Autonomous) exceeded 80.00%. The cumulative variance contribution rate of the first principal component was 61.18%, whereas that of the second principal component was 22.78%. One of the Dorbod Mongol Autonomous samples was distinguished from Tailai (Figure 7b). Upon synthesis, it becomes evident that there is a crossover between the two origins of the smaller traceability scale (Tailai-Baicheng and Dorbod Mongol Autonomous). The traceability range of the collected samples had a significant impact on the origin discrimination results. Consequently, further KNN analysis was conducted on the PCA results of the near-infrared (NIR) spectral data with varying sample traceability scales.

#### 3.3.2. Effect of Traceability Scale on Origin Discrimination Results Based on KNN Analysis

A near-infrared discrimination model of mung bean samples with different sample traceability scales was established by combining the model with the KNN method, based on PCA feature extraction after spectral preprocessing of mung bean samples at larger (Tailai-Sishui) and smaller scales (Tailai-Baicheng and Tailai-Dorbod Mongol Autonomous). The same 3.1.3 method yielded more expansive and condensed scales of mung bean origin discrimination classification outcomes. As illustrated in Table 8 and Figure 8, the overall accuracy of the origin discrimination at the larger scale (A) was 100.00%; the overall accuracy of the origin discrimination at the smaller scale (B) exhibited a 1.48% improvement. A comparison of the results of discriminating the origins of mung beans at different traceability scales indicated that the model established using the larger scale of mung bean origin samples exhibited a higher correct rate of discrimination than the model established using the smaller scale of mung bean origin samples.

He et al. conducted a comparative analysis of the mineral element content in Maca sourced from large-scale (Peru and China) and small-scale regions (various Chinese provinces, including Yunnan, Xinjiang, and Tibet) using ICP-MS. The overall correct discrimination rate of Maca samples from large-scale regions (across different countries) exceeded 95.00% when PCA and discriminant analysis were used, whereas the overall correct discrimination rate of Maca samples from small-scale regions (across different provinces) surpassed 80.00% [28]. This indicates that the geographical distance of the sample source influences the efficacy of discrimination, with a greater distance resulting in a higher rate of discrimination accuracy.

To assess the discriminant models constructed from the NIR spectral data of varying sample traceability scales, the evaluation indices of the discriminant models for the larger and smaller scales of mung bean origins are presented in Table 9. As illustrated in Table 9, the accuracy of the origin traceability discrimination model established based on mung bean samples from a larger scale (Tailai-Sishui) was 98.00%, with an F_1_-score of 1.00, which was nearly identical to the correct discrimination rate of origin discrimination. In contrast, the accuracy of the origin traceability discrimination model based on mung bean samples from smaller scales (Tailai-Baicheng, Tailai-Dorbod Mongol Autonomous) was enhanced by 1.75%, with an F_1_-score of 0.20. A comparison of the evaluation indicators of the mung bean origin discrimination model for larger and smaller scales reveals that, for smaller scales of the origin, the discrimination effect is slightly lower than that for larger scales. This suggests that the scale of sample traceability is a crucial factor affecting the development of discriminant models. Accurate discrimination of traceability models based on small areas is challenging, likely because of the influence of the traceability of agricultural products in the surrounding area.

### 3.4. Effect of Shelf Life on the Accuracy of the Near-Infrared Origin Traceability of Mung Beans

The availability of mung bean samples of different origins at different times may cause fluctuations in the organic components (protein, starch, fat, etc. [29]) that are present in these beans. This is due to several factors, including climatic conditions, soil type, cultivation methods, and varietal differences [30]. Over time, these factors may affect the organic content of mung beans [31]. To date, few studies have examined the impact of shelf life of different origins on the efficacy of traceability discrimination models for agricultural products based on NIRS technology. Therefore, the impact of the shelf life of mung beans of different origins on the ability of NIR spectroscopy to discriminate between their origins was investigated.

#### 3.4.1. Effect of Sample Shelf Life on Origin Discrimination Results Based on PCA

As illustrated in Figure 9, the sum of the contribution rates of the first and second PCs in four origins at different shelf lives (90, 180, 270, and 360 d) exceeded 90.00%, with the first principal component contributing 81.00% of the cumulative variance and the second principal component contributing 10.04% of the cumulative variance. However, by synthesizing, it can be visualized that there is a crossover between the four origins for the different shelf lives. This demonstrated that the disparate shelf lives of the collected samples influenced the outcomes of the origin discrimination.

#### 3.4.2. Effect of Sample Shelf Life on Origin Discrimination Results Based on KNN Analysis

A near-infrared discrimination model of mung bean samples with different shelf lives (90, 180, 270, and 360 d) was established based on the PCA feature extraction information and spectral preprocessing of the samples. The KNN method was used to combine the data and establish the model. The same 3.1.3 method was used to obtain the classification results for mung bean origin discrimination regarding different shelf lives (Table 10). As illustrated in Table 10, the overall accuracies of the origin discrimination for samples with different shelf lives (90, 180, 270, 360, 90–180–270–360 d) were 86.65%, 91.70%, 92.96%, 93.85%, and 94.50% for the four origins (90–180–270–360 d > 360 d > 270 d > 180 d > 90 d).

The origin discrimination of mung beans obtained from the four origins with a shelf life of 90 d included 135 samples that were correctly distinguished and 23 samples that were incorrectly distinguished. The origin discrimination of mung beans obtained from the four origins with a shelf life of 180 d included 143 samples that were correctly distinguished and 13 samples that were incorrectly distinguished. The origin discrimination of mung beans obtained from the four origins with a shelf life of 270 d included 131 samples that were correctly distinguished and 10 samples that were incorrectly distinguished. The origin discrimination of mung beans obtained from the four origins with a shelf life of 360 d included 137 samples that were correctly distinguished and 9 samples that were incorrectly distinguished. The origin discrimination of mung beans obtained from the four origins with a shelf life of 90–180–270–360 d included 568 samples that were correctly distinguished and 32 samples that were incorrectly distinguished. A comparison of the results of the mung bean origin discrimination based on different shelf lives indicated that the correct discrimination rate of the model developed for mung bean origin samples with multiple shelf lives (90–180–270–360 d) was higher than the model developed for mung bean origin samples with a single shelf life (90 d).

In a study conducted by Liu et al., NIRS was used to collect citrus samples with varying shelf lives (10, 20, and 30 d). The correct discrimination rate of citrus samples with shelf lives of 10, 20, and 30 d was 80.00% [32]. Similarly, Li et al. used electric rate neural networks to classify egg storage periods and achieved a discrimination accuracy of 92.86% for egg samples with storage periods ranging from 0 to 180 d [33]. Zhang et al. used NIRS to differentiate the shelf lives of diverse apple varieties. Following preprocessing techniques such as Detrending and SNV, the accuracy of discerning the shelf life of their apple samples (0, 14, and 28 d) was 91.67%, 95.00%, and 96.67%, respectively [34]. This indicates that the selection of shelf life may influence the ability to discriminate between the origins of mung beans.

To validate the comparison of the discriminant model’s accuracy with multiple shelf lives (90–180–270–360 d) and a single shelf life (90 d, 180 d, 270 d, 360 d), five samples from outside the modeling of the four origins were selected for substitution into the discriminant model with different shelf lives. The results of the mung bean origin discriminant model with different shelf lives for the external validation of the discriminant model outside the collection model are presented in Table 11.

As illustrated in Table 11, twenty validation samples were compared using the discrimination model developed for the four origins with a shelf life of 90 d. Eight samples were misidentified. Specifically, two Tailai and one Dorbod Mongol Autonomous samples were misclassified as Sishui, three Baicheng samples were misclassified as Tailai and Dorbod Mongol Autonomous, and two Sishui samples were misclassified as Baicheng. Twenty validation samples were compared using the discrimination model developed for the four origins with a shelf life of 180 d. Eight samples were misidentified. Specifically, two Tailai samples were misclassified as Sishui, two Dorbod Mongol Autonomous samples were misclassified as Baicheng, two Baicheng samples were misclassified as Dorbod Mongol Autonomous, and two Sishui samples were misclassified as Dorbod Mongol Autonomous and Baicheng. Twenty validation samples were compared using the discrimination model developed for the four origins with a shelf life of 270 d. Seven samples were misidentified. Specifically, two Tailai samples were misclassified as Sishui, one Dorbod Mongol Autonomous sample was misclassified as Baicheng, two Baicheng samples were misclassified as Dorbod Mongol Autonomous, and two Sishui samples were misclassified as Tailai and Baicheng. Twenty validation samples were compared using the discrimination model developed for the four origins with a shelf life of 360 d. Six samples were misidentified, with one sample from Tailai being misclassified as Dorbod Mongol Autonomous, one sample from Dorbod Mongol Autonomous being misclassified as Baicheng, two samples from Baicheng being misclassified as Dorbod Mongol Autonomous, and two samples from Sishui being misclassified as Tailai.

Twenty validation samples were then compared using the discriminant model developed for the four origins with shelf lives of 90–180–270–360 d. Four samples were misidentified, with one sample from Dorbod Mongol Autonomous being misclassified as Tailai, two samples from Baicheng being misclassified as Tailai and Sishui, and one sample from Sishui being misclassified as Tailai. This indicates that for mung bean samples with multiple shelf lives (90, 180, 270, and 360 d), the discriminant model developed for multiple shelf lives (90, 180, 270, and 360 d) was more accurate than that developed for a single shelf life (90 d). A comparison of the evaluation indices of the mung bean origin discrimination model with multiple and single shelf lives revealed that the greater the number of shelf lives there are in the model, the more effective the discrimination based on different mung bean origins is.

## 4. Conclusions

In this study, spectral acquisition of mung bean samples of different origins was performed using NIRS. The objective of this study was to investigate the use of PCs to analyze near-infrared spectral data of mung beans of different origins after spectral preprocessing of mung bean samples that had undergone different spectral preprocessing methods. The results indicate that the total cumulative variance contribution rate of the first three PCs was 98.16%. It represented most of the near-infrared spectral information that was present in the mung beans of different origins. Furthermore, the accuracy of the mung bean origin discrimination model established by combining the model with the KNN method was 98.25%. This demonstrates the feasibility of using NIRS with the KNN method to conduct discriminative studies on mung beans of varying origins. Nevertheless, the results are impacted by numerous variables, including the sample quantity, sample traceability scale, and shelf life.

The impacts of sample quantity, traceability scale, and shelf life on the NIR traceability discrimination results of mung beans of different origins were investigated. (a) The origin discrimination model exhibited greater accuracy with a larger sample quantity (600) than with a smaller sample quantity (200). The accuracy of the origin discrimination model can be enhanced by selecting the largest feasible quantity of samples as an indicator of the accuracy of origin discrimination. (b) The accuracy of the origin traceability discrimination model established for mung bean samples at larger traceability scales (Tailai-Sishui) versus smaller traceability scales (Tailai-Baicheng, Tailai-Dorbod Mongol Autonomous) was enhanced, whereas the accuracy of the model discrimination based on small scales was lower. (c) The original discrimination model based on multiple shelf lives (90–180–270–360 d) demonstrated greater accuracy than that based on a single shelf life (90, 180, 270, 360 d). These findings demonstrate that shelf life is a crucial factor that influences the development of an effective origin discrimination model. This model can effectively discriminate mung bean origin traceability and provide novel insights and research avenues for the study of the traceability of agricultural products.

Although this study investigated the influence of different geographical origins of the mung bean sample traceability scale, shelf life, and other factors on the model discrimination results, studying the sample traceability scale remains challenging at a limited scale of origin traceability and accurate discrimination. Due to the large differences in the origin, climate, soil, water, and other factors among a large amount of samples [35,36], the annual average temperature and soil types of the three geographical origins in Northeast (Jilin, Tailai, and Dorbod Mongol Autonomous) are similar (4.5–6.0 °C (2022–2023), including caltunica nigritesoil, meadow soil, and sand soil), while the annual average temperature in Shandong (Sishui) is 13.4 °C (2022–2023), and the soil types are brown loam and brown soil. This is also the main reason for the small differences in near-infrared feature information from the small-scale geographical origins in Northeast. Kaoru et al. used high-resolution inductively coupled plasma mass spectrometry to determine the mineral element contents in rice from a large area (Japan, Thailand, the United States, China). The results showed that the discrimination accuracy rate of the origin traceability model based on different origins was 97.0% [37]. Shi et al. used mineral element fingerprint technology to determine the mineral element contents in rice from small areas (Songjiang and Jinshan and Chongming), and the results showed that the discrimination accuracy of the origin traceability model based on different origins was 92.1% [38]. The above studies’ conclusions are consistent with the conclusions of this study. Therefore, further supplementation with different agricultural products from other regions is necessary to meet the requirements of the sample traceability scale, which affects the origin traceability identification for different agricultural product research.

Furthermore, this study demonstrated a correlation between the overall correct discrimination rate of mung bean origins from different regions and a discrimination model based on the shelf life of different samples. In the process of sample discrimination, over time, aspects such as the moisture and protein in the sample may undergo slow changes [39,40], which have an impact on the collection of near-infrared spectroscopy data and the discrimination results of the established model. Even within the same shelf life, the storage conditions of samples of different origins (such as humidity, temperature, light, etc.) may have different effects on the model’s discrimination performance. Meanwhile, samples of different origins may have varying abilities to trace and distinguish the origin of different agricultural products due to differences in natural and anthropogenic environments. Consequently, it is essential to consider the impact of a sample’s shelf life on the development of future standard system technologies for the traceability of agricultural products. The sample collection process for research on the traceability of agricultural products should aim to include numerous samples with different shelf lives to enhance the stability and accuracy of the discriminative model.

## Figures and Tables

**Figure 1 foods-13-03234-f001:**
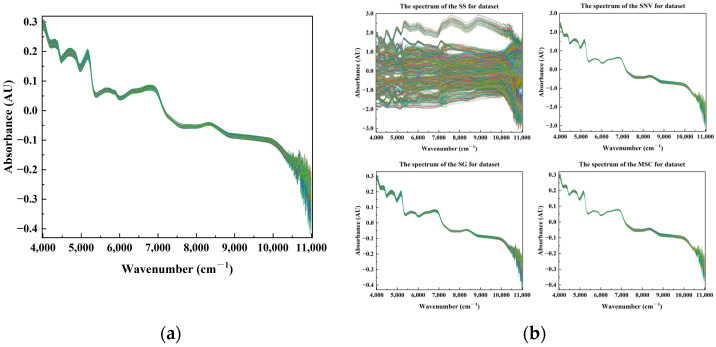
Near-infrared raw spectra of mung beans of different origins and their near-infrared spectra after processing by preprocessing methods: (**a**) near-infrared raw spectra of mung beans of different origins; (**b**) near-infrared spectra of mung bean after processing using six spectral preprocessing methods such as Standardization (SS), Savitzky–Golay (SG), Standard Normal Variable (SNV), and Multiple Scattering Correction (MSC), respectively. (The lines represent a different spectral absorption peak or characteristic absorption band).

**Figure 2 foods-13-03234-f002:**
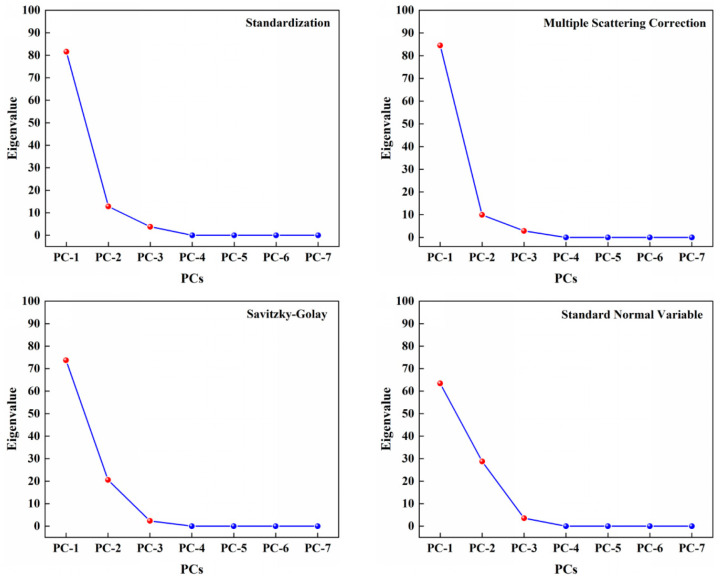
Principal component rubble map of mung beans of different origins.

**Figure 3 foods-13-03234-f003:**
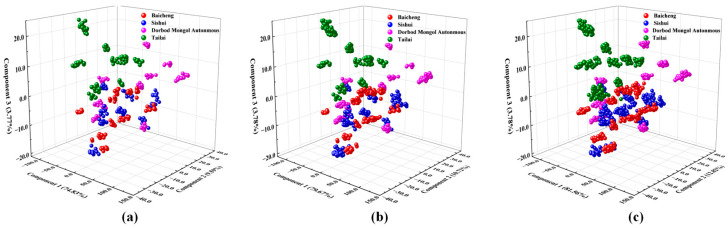
Distribution map of mung bean feature vectors of the top 3 components extracted from PCA features with different sample quantities: (**a**) principal component plots of mung beans of different origins for 200; (**b**) principal component plots of mung beans of different origins for 400; (**c**) principal component plots of mung beans of different origins for 600.

**Figure 4 foods-13-03234-f004:**
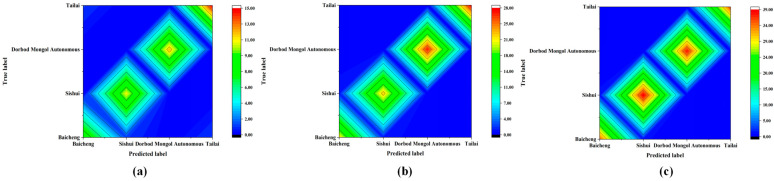
Contour plot of mung bean origin discrimination for different sample quantities: (**a**) contour map of different mung bean origins discriminated using a sample size of 200; (**b**) contour map of different mung bean origins discriminated using a sample size of 400; (**c**) contour map of different mung bean origins discriminated using a sample size of 600.

**Figure 5 foods-13-03234-f005:**
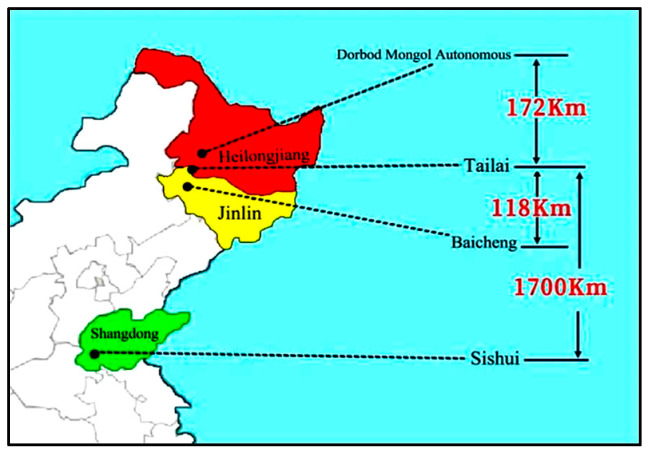
Plot of actual distance of origins of different categories of mung bean samples.

**Figure 6 foods-13-03234-f006:**
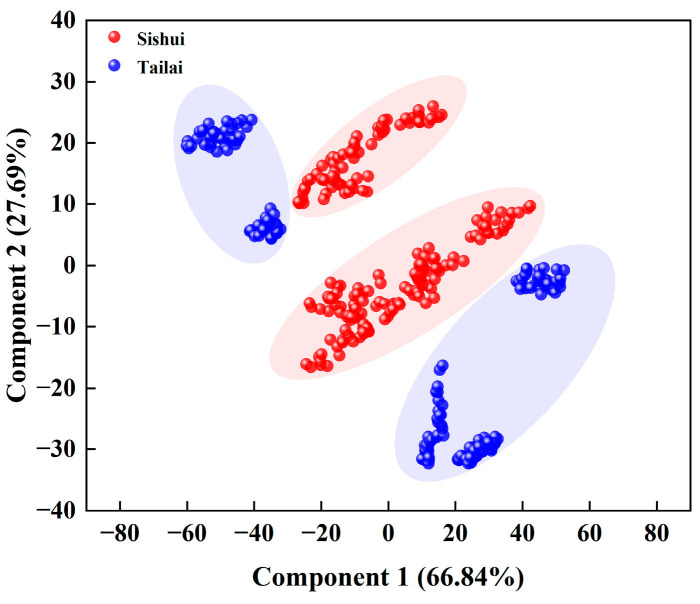
PCA plot of mung bean discrimination at a larger scale.

**Figure 7 foods-13-03234-f007:**
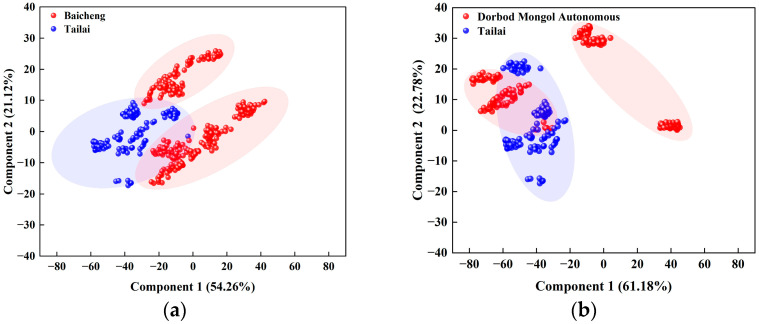
PCA plot of mung bean discrimination at a smaller scale: (**a**) principal component plot of mung bean at the scale of Tailai-Baicheng; (**b**) principal component plot of mung bean at the scale of Tailai-Dorbod Mongol Autonomous.

**Figure 8 foods-13-03234-f008:**
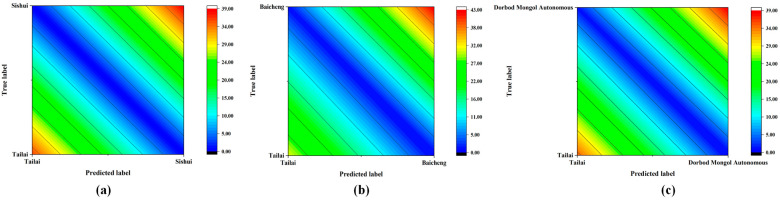
Contour plot of mung bean origin discrimination for larger and smaller scales: (**a**) contour map of mung bean at the scale of Tailai-Sishui; (**b**) contour map of mung bean at the scale of Tailai-Baicheng; (**c**) contour map of mung bean at the scale of Tailai-Borbod Mongol Autonomous.

**Figure 9 foods-13-03234-f009:**
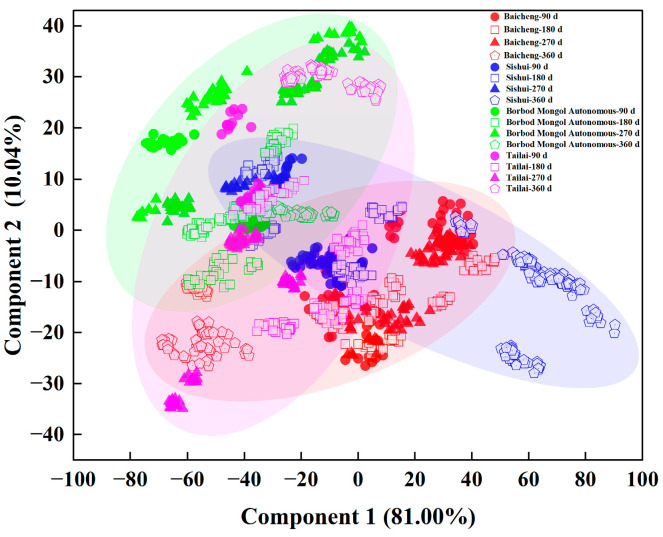
PCA plot of mung bean discrimination based on shelf life.

**Table 1 foods-13-03234-t001:** The sample numbers and location of mung beans’ geographical origins.

Region	Number of Samples	Variety	E Longitude (deg)	N Latitude (deg)
Baicheng, Jilin Province	30	Polo green, pearl green, hairy mung beans	122˚38′–123˚22′	45˚13′–45˚18′
Sishui County, Shandong Province	30	Parrot, hairy mung beans	114°48′–116°24′	34°39′–35°52′
Dorbod Mongol Autonomous County, Heilongjiang Province	30	Bright, yellow, hairy, middle, parrot mung beans	124°44′–124°46′	46°86′–46°96′
Tailai County, Heilongjiang Province	30	Polo green, bright mung beans	123°42′–124˚20′	46˚39′–46˚59′

**Table 2 foods-13-03234-t002:** The cumulative variance contributions of the various spectral preprocessing methods to each principal component (PC) of the NIR spectra.

Preprocessing Methods	Component 1 (%)	Component 2 (%)	Component 3 (%)	Total (%)
Standardization (SS)	81.56	12.82	3.78	98.16
Multiple Scattering Correction (MSC)	84.48	9.93	2.89	97.31
Savitzky–Golay (SG)	73.73	20.54	2.33	96.60
Standard Normal Variable (SNV)	63.42	28.71	3.54	95.67

**Table 3 foods-13-03234-t003:** Classification results of near-infrared spectral origin discrimination of mung beans of different origins.

Origin Classification	Number of Correct Classifications	Number of Incorrect Classifications	Accuracy (%)	Total
Baicheng	35	1	97.22	36
Sishui	39	1	97.50	40
Dorbod Mongol Autonomous	38	0	100.00	38
Tailai	36	0	100.00	36

**Table 4 foods-13-03234-t004:** Evaluation indexes of near-infrared discrimination model for mung beans of different origins.

Origin Classification	Precision (%)	Recall (%)	F_1_-Score	Accuracy (%)
Baicheng	96.00	96.00	0.97	98.25
Sishui	97.00	98.00	0.97	
Dorbod Mongol Autonomous	100.00	100.00	1.00	
Tailai	100.00	100.00	1.00	

**Table 5 foods-13-03234-t005:** The cumulative variance contributions of the different sample quantities to each principal component of the mung bean NIR spectra.

Sample Quantity	Component 1 (%)	Component 2 (%)	Component 3 (%)	Total (%)
A	74.83	9.59	3.77	88.19 ^c^
B	79.67	10.72	3.78	94.17 ^b^
C	81.56	12.82	3.78	98.16 ^a^

^a–c^ in the same column indicate that there are significant differences among sample quantities at *p* < 0.05 level.

**Table 6 foods-13-03234-t006:** Classification results of different sample quantities for mung bean origin discrimination.

Sample Quantity	Origin Classification	Number of Correct Classifications	Number of Incorrect Classifications	Accuracy (%)	Total
A	Baicheng	9	1	90.00	10
	Sishui	11	1	91.67	12
	Dorbod Mongol Autonomous	12	0	100.00	12
	Tailai	15	1	93.75	16
B	Baicheng	21	1	95.45	22
	Sishui	22	1	95.65	23
	Dorbod Mongol Autonomous	27	0	100.00	27
	Tailai	28	0	100.00	28
C	Baicheng	35	1	97.22	36
	Sishui	39	1	97.50	40
	Dorbod Mongol Autonomous	38	0	100.00	38
	Tailai	36	0	100.00	36

**Table 7 foods-13-03234-t007:** Evaluation indexes of near-infrared discriminant model for mung beans using different sample quantities.

Sample Quantity	Origin Classification	Precision (%)	Recall (%)	F_1_-Score	Accuracy (%)
200 (A)	Baicheng	89.00	90.00	0.90	93.00
	Sishui	91.00	92.00	0.90	
	Dorbod Mongol Autonomous	100.00	100.00	1.00	
	Tailai	92.00	93.00	0.93	
400 (B)	Baicheng	93.00	94.00	0.92	96.75
	Sishui	94.00	94.00	0.94	
	Dorbod Mongol Autonomous	100.00	100.00	1.00	
	Tailai	100.00	100.00	1.00	
600 (C)	Baicheng	96.00	96.00	0.97	98.25
	Sishui	97.00	98.00	0.97	
	Dorbod Mongol Autonomous	100.00	100.00	1.00	
	Tailai	100.00	100.00	1.00	

**Table 8 foods-13-03234-t008:** Classification results of mung bean origin discrimination at larger and smaller scales.

Traceability Scale	Origin Classification	Number of Correct Classifications	Number of Incorrect Classifications	Accuracy (%)	Total
Larger scale (A)	Tailai	36	0	100.00	36
	Sishui	39	0	100.00	39
Smaller scale (B)	Tailai	31	1	96.88	32
	Baicheng	43	0	100.00	43
	Tailai	35	1	97.22	36
	Dorbod Mongol Autonomous	39	0	100.00	39

**Table 9 foods-13-03234-t009:** Indicators for evaluating models for discriminating between larger- and smaller-scale mung bean origins.

Traceability Scale	Origin Classification	Precision (%)	Recall (%)	F_1_-Score	Accuracy (%)
Larger scale (A)	Tailai	100.00	100.00	1.00	100.00
	Sishui	100.00	100.00	1.00	
Smaller scale (B)	Tailai	100.00	99.00	0.99	98.50
	Baicheng	97.00	100.00	0.99	
	Tailai	100.00	98.00	0.99	98.00
	Dorbod Mongol Autonomous	96.00	100.00	0.98	

**Table 10 foods-13-03234-t010:** Classification results of origin discrimination of mung beans with different shelf lives.

Origin Classification	Shelf Life	Number of Correct Classifications	Number of Incorrect Classifications	Accuracy (%)	Total
Baicheng	T_1_	39	13	75.00	52
	T_2_	34	3	91.89	37
	T_3_	29	2	93.55	31
	T_4_	36	2	94.74	38
	T_1_–T_2_–T_3_–T_4_	151	7	95.57	158
Sishui	T_1_	31	4	88.57	35
	T_2_	38	4	90.48	42
	T_3_	33	3	91.67	36
	T_4_	41	3	93.18	44
	T_1_–T_2_–T_3_–T_4_	146	10	93.59	156
Dorbod Mongol Autonomous	T_1_	35	3	92.11	38
	T_2_	36	3	92.31	39
	T_3_	30	2	93.75	32
	T_4_	31	2	93.94	33
	T_1_–T_2_–T_3_–T_4_	134	8	94.37	142
Tailai	T_1_	30	3	90.91	33
	T_2_	35	3	92.11	38
	T_3_	39	3	92.86	42
	T_4_	29	2	93.55	31
	T_1_–T_2_–T_3_–T_4_	137	8	94.48	145

**Table 11 foods-13-03234-t011:** Validation results of mung bean origin discrimination models with different shelf lives on the collection of results in extra-discriminative models.

Shelf Life	Random Samples (Number)	Origin Classification	Number of Discriminations (Baicheng)	Number of Discriminations (Sishui)	Number of Discriminations (Dorbod Mongol Autonomous)	Number of Discriminations (Tailai)	Accuracy (%)
90 d	20	Baicheng (5)	2	0	2	1	60.00
		Sishui (5)	2	3	0	0	
		Dorbod Mongol Autonomous (5)	0	1	4	0	
		Tailai (5)	0	2	0	3	
180 d	20	Baicheng (5)	3	0	2	0	60.00
		Sishui (5)	1	3	1	0	
		Dorbod Mongol Autonomous (5)	2	0	3	0	
		Tailai (5)	0	2	0	3	
270 d	20	Baicheng (5)	3	0	2	0	65.00
		Sishui (5)	1	3	0	1	
		Dorbod Mongol Autonomous (5)	1	0	4	0	
		Tailai (5)	0	2	0	3	
360 d	20	Baicheng (5)	3	0	2	0	70.00
		Sishui (5)	0	3	0	2	
		Dorbod Mongol Autonomous (5)	1	0	4	0	
		Tailai (5)	0	0	1	4	
90–180–270–360 d	20	Baicheng (5)	3	1	0	1	80.00
		Sishui (5)	0	4	0	1	
		Dorbod Mongol Autonomous (5)	0	0	4	1	
		Tailai (5)	0	0	0	5	

## Data Availability

The original contributions presented in the study are included in the article, further inquiries can be directed to the corresponding author.

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
