# Peer review of "The Impact of Sample Quantity, Traceability Scale, and Shelf Life on the Determination of the Near-Infrared Origin Traceability of Mung Beans"

_foods, 2024, doi:10.3390/foods13203234_

Round 1
Reviewer 1 Report
Comments and Suggestions for Authors
A very well written paper, congratulations to authors.
The manuscript Sample quantity, traceability scope, and shelf life on the discrimination of near-infrared origin traceability of mung beans, authors Ming-Ming Chen, Yan Song, Yan-Long Li, Xin-Yue Sun, Feng Zuo, Li-Li Qian, address the understanding the impact of sample quantity, traceability range, and shelf life on the accuracy of mung bean origin traceability models using near-infrared spectroscopy (NIRS). Multivariate statistics based on multivariate analysis was used to find the hidden relationships between variables. The topic is significant and represents a step forward in the research in this field since it combines analytical results with advanced statistics to find new possibilities to ensure food traceability. The model was applied to mung beans, but can be extended to other foods.
The introduction section is well written, clearly presenting the state of the art and what this paper brins new. The results were obtained in a well-organized methodology. Sample collection, pretreatment, analytical methods, and statistical approaches are well described. Data are presented in a readable manner, and compared with other previous studies. The figures are representative and of good quality. The number of tables is adequate. Conclusions are supported by obtained results. The references are up to date and appropriate.
Author Response
Comments 1: The manuscript Sample quantity, traceability scope, and shelf life on the discrimination of near-infrared origin traceability of mung beans, authors Ming-Ming Chen, Yan Song, Yan-Long Li, Xin-Yue Sun, Feng Zuo, Li-Li Qian, address the understanding the impact of sample quantity, traceability range, and shelf life on the accuracy of mung bean origin traceability models using near-infrared spectroscopy (NIRS). Multivariate statistics based on multivariate analysis was used to find the hidden relationships between variables. The topic is significant and represents a step forward in the research in this field since it combines analytical results with advanced statistics to find new possibilities to ensure food traceability. The model was applied to mung beans, but can be extended to other foods.
Response 1: Thanks for your comments and recognition. As you mentioned, this model method is also applicable to other agricultural products. It is feasible to study the traceability and discrimination of mung beans from different origins by using near-infrared spectroscopy technology. However, due to factors such as sample quantity, traceability range, and shelf life, this applies not only to mung bean samples with geographical indications, but also to other agricultural products.
Comments 1: The introduction section is well written, clearly presenting the state of the art and what this paper brins new. The results were obtained in a well-organized methodology. Sample collection, pretreatment, analytical methods, and statistical approaches are well described. Data are presented in a readable manner, and compared with other previous studies. The figures are representative and of good quality. The number of tables is adequate. Conclusions are supported by obtained results. The references are up to date and appropriate.
Response 1: Thanks for your affirmation. From the results of this study, it appears to be quite ideal. The near-infrared spectroscopy traceability technology is indeed affected by factors such as sample quantity, traceability range, and shelf life. Of course, there are other factors that are also affecting its research results, and we will continue to conduct in-depth systematic research in the future.

Reviewer 2 Report
Comments and Suggestions for Authors
In the manuscript “Sample quantity, traceability scope, and shelf life on the discrimination of near-infrared origin traceability of mung beans” the authors aim to address the gap in understanding the impact of sample quantity, traceability range, and shelf life on the accuracy of mung bean origin traceability models based on near infrared spectroscopy. The approach is interesting but manuscript has several gaps that need to be improved prior the publication. My suggestion is major revision according to following comments:
Abstract should include information about NIR range.
Abstracts should also include information about used pre-processing methods.
The introduction discusses several studies that used NIRS for traceability, but there is a lack of explanation of the underlying mechanisms that make NIRS suitable for this purpose. Why is NIRS effective for tracing geographical origins?
Many of the studies cited emphasize high accuracy in discriminating origins. However, the introduction lacks a critical discussion of the limitations or challenges these models might face, such as overfitting, model robustness, or potential variability under different conditions (e.g., different growing seasons or environmental factors).
The introduction mentions the successful application of NIR, but there is little discussion of the technological limitations of NIR spectroscopy. For example, how does the quality of the NIR equipment, calibration issues, or environmental interference (e.g., humidity, temperature) affect the accuracy of these models? A gap in addressing these technological limitations could be critical for real-world implementation.
The novelty of the research is not clear from the introduction section.
The sampling points in each region are described as east, west, north, south, and center, but there is no explanation for why these points were chosen. Was the checkerboard sampling method optimal for the specific geographic and environmental characteristics of the regions? More detail on how this method ensures representativeness would be helpful.
Although it mentions that mung beans were collected from different regions, there is no mention of whether different varieties of mung beans were considered. This is a gap because mung bean varieties may affect the spectral characteristics, potentially impacting the model's accuracy.
The experiment does not mention any consideration of environmental factors (e.g., soil composition, weather patterns, irrigation methods) in the different regions. These factors could affect the chemical makeup of the mung beans, which might influence the NIR spectra. Their absence may lead to oversights in interpreting the spectral differences purely based on geographical origin.
Line 141. How were grinded samples stored?
Line 154. How were the “most optimal spectra” selected?
Several preprocessing techniques are used (e.g., Standardization, Savitzky-Golay, SNV, MSC), but there is no rationale provided for why these specific methods were chosen or how they improve the results. It's unclear whether other techniques were considered or why the listed methods were deemed optimal for mung beans.
The experimental section discusses creating discriminant models using PCA and KNN but does not mention how the model performance was evaluated beyond dividing data into training and validation sets. Details about key metrics like accuracy, precision, recall, F1 score, or confusion matrices are missing. These metrics are crucial to assessing model robustness and generalizability.
A basic split into training and validation sets is mentioned, but cross-validation (such as k-fold) could improve the rigor of model evaluation. Not using cross-validation can result in overfitting, especially with smaller datasets.
The sample sizes for PCA modeling (n=200, 400, 600) are mentioned, but there's no explanation for why these specific numbers were chosen. Was there an analysis to ensure these numbers were statistically significant for the geographic regions studied? Additionally, no randomization or repetition of sampling is mentioned, which could introduce sampling bias.
The model's performance with different sample sizes (200, 400, 600) is only assessed in a specific geographic scope. It would be important to test the model on a more diverse set of regions or agricultural products to determine its broader applicability.
Line 241. The text mentions using the KNN method but does not specify the number of neighbors (k) used in the analysis or any other parameters. Providing this detail would help in understanding how the model's performance was achieved. While K-fold cross-validation is mentioned, it would be useful to know how many folds were used, and if any other validation techniques were considered or used.
Line 281. There’s mention of PCA results but not a clear connection on how each preprocessing method influenced the PCA results. A more detailed analysis of how each preprocessing method affected the PCA output could provide deeper insights.
Please offere more in-depth statistical analysis and significance testing. The differences in accuracy for various sample sizes are presented, but there is no mention of statistical tests to determine whether these differences are significant. This would be important for evaluating the reliability of the results. Were any statistical tests (e.g., t-tests, ANOVA) performed to determine if the accuracy differences between sample sizes are statistically significant?
Expanding the comparison with other studies. The text briefly mentions studies by Li et al. and Qiu et al. but does not explain how these studies are directly comparable or what lessons can be drawn from their methodologies. Further discussion on differences in methodologies and datasets would add depth to the comparison.
The text does not mention the computational requirements for processing and modeling with different sample sizes. This is important for other researchers who might want to replicate the study, especially when handling large datasets.
The findings are specific to mung beans, but there is no discussion about whether the results can be applied to other agricultural products. A broader discussion about the potential generalizability of the results would enhance the impact of the study.
References should be revised. References should include abbreviated journal titles.
Author Response
Comments 1: Abstract should include information about NIR range.
Response 1: Thanks for the suggestion. We added the information about NIR range (Line 17).
Comments 2: Abstracts should also include information about used pre-processing methods.
Response 2: Thanks for the suggestion and we added the information about used pre-processing methods (Line 18-19).
Comments 3: The introduction discusses several studies that used NIRS for traceability, but there is a lack of explanation of the underlying mechanisms that make NIRS suitable for this purpose. Why is NIRS effective for tracing geographical origins?
Response 3: Thanks for the suggestion and we added the sentence: NIRS utilizes specific functional groups (such as C-H, O-H, N-H) in substances to selectively absorb near-infrared light. Different geographical locations and growth conditions can affect the distribution pattern of these functional groups in substances, forming unique spectral features. By analyzing these spectra, specific chemical compositions of different regions can be reflected, forming the theoretical basis for near-infrared spectroscopy for origin tracing (Line 74-80).
Comments 4: Many of the studies cited emphasize high accuracy in discriminating origins. However, the introduction lacks a critical discussion of the limitations or challenges these models might face, such as overfitting, model robustness, or potential variability under different conditions (e.g., different growing seasons or environmental factors).
Response 4: Thanks for the suggestion and we added the discussion of the limitations or challenges these models might face. the application prospects of NIR spectral analysis models in the field of agricultural product origin tracing are broad, but they also face challenges such as overfitting, model robustness, and potential variability under different conditions: (a) Overfitting is one of the common problems in machine learning. In NIR spectral analysis, overfitting may be caused by various factors (insufficient training data, excessive or underfitting model parameters, etc); (b) Model robustness may be affected by various factors (sensitivity to differences in sample type, composition, and structure), which may lead to a decrease in the predictive accuracy of the model. (c) Potential variability under different conditions (growing seasons or environmental factors), the growth rate and quality of the sample may change, which will affect the accuracy of the NIR spectral analysis model. The model needs to be able to adapt to these changes in order to maintain high prediction accuracy under different conditions (Line 94-105).
Comments 5: The introduction mentions the successful application of NIR, but there is little discussion of the technological limitations of NIR spectroscopy. For example, how does the quality of the NIR equipment, calibration issues, or environmental interference (e.g., humidity, temperature) affect the accuracy of these models? A gap in addressing these technological limitations could be critical for real-world implementation.
Response 5: Thanks for the suggestion and we added the discussion about the quality of the NIR equipment, calibration issues, or environmental interference (e.g., humidity, temperature) affect the accuracy of these models. (a) Equipment quality: If the light source of NIR equipment is unstable, the intensity and wavelength of the emitted near-infrared light may fluctuate, making it difficult to accurately measure the absorption and reflection characteristics of the sample to light; (b) Calibration issues: When analyzing a certain chemical substance, wavelength calibration errors can cause the position of characteristic absorption peaks to shift, and the model may not be able to correctly identify them; (c) Environmental interference (e.g., humidity, temperature): When the humidity is high, the surface of optical components (lenses, mirrors, etc.) in NIR devices may adsorb water vapor, causing the focusing position of light passing through the lens to shift, resulting in changes in the interaction between light and the sample, thereby affecting the collection of spectral data; When the temperature rises, it may cause inaccurate focusing of light, affecting the accuracy of spectral measurements and ultimately reducing the accuracy of the model (Line 82-94).
Comments 6: The novelty of the research is not clear from the introduction section.
Response 6: Thanks for the suggestion. Previous studies have demonstrated that different types of agricultural products analyzed using near-infrared spectroscopy exhibit variations in chemical composition and physical properties, which can be attributed to various factors, including origin, variety, year, and processing technology. An appropriate model was established using the characteristic information obtained by near-infrared spectroscopy, and the impact of these factors on the results of the discernment of the origin was analyzed. However, there is a paucity of literature on the effects of the distance range of the sample origin, sample quantity collected, and shelf life of the samples on the discrimination results of the origin traceability. Consequently, it can be regarded as a pivotal aspect of the investigation into the influence of traceability technology on agricultural products, and its variables can be evaluated for their impact on the original traceability model. Consequently, it can be regarded as a pivotal aspect of the investigation into the influence of traceability technology on agricultural products, and analyze the impact of its factors on the origin traceability model as a research innovation point (Line 128-138).
Therefore, In this study, we analyzed the results of near-infrared spectroscopy (NIRS) to discriminate between mung bean origins. To confirm the feasibility of tracing mung bean samples based on NIR spectroscopy. Furthermore, the impact of sample quantity, traceability range, and shelf life on the establishment of different mung bean origin traceability discriminant models was analyzed.
Comments 7: The sampling points in each region are described as east, west, north, south, and center, but there is no explanation for why these points were chosen. Was the checkerboard sampling method optimal for the specific geographic and environmental characteristics of the regions? More detail on how this method ensures representativeness would be helpful.
Response 7: Thanks for the suggestion. The commonly used sampling methods include five point sampling, diagonal sampling, chessboard sampling, parallel line sampling, “Z”-shaped sampling, etc.. In this study, to ensure the representativeness and uniformity of the collected samples. Therefore, a chessboard sampling method was adopted, and sampling locations were set according to the different production areas, regions, and planting ranges, forming a chessboard pattern. Then, the sampling locations were evenly distributed on certain blocks of the field. Randomly set 5 duplicate sampling points within each region, and the sampling locations were established at five strategic points-east, west, south, north, and center-within each region. (https://doi.org/10.3969/j.issn.1002-2090.2023.01.010).
Comments 8: Although it mentions that mung beans were collected from different regions, there is no mention of whether different varieties of mung beans were considered. This is a gap because mung bean varieties may affect the spectral characteristics, potentially impacting the model's accuracy.
Response 8: Thanks for the suggestion. When conducting traceability discrimination of mung beans from different origins in this study, without considering the variety, they have different characteristics from different origins. Even if they are from different origins under the same variety, they can be classified and identified well. For example, both Baicheng and Tailai have the same variety (Polo green), but their production areas still have a significant impact on them. In the future, we will also consider studying the impact of varieties, origins, and their interactions on near-infrared spectra as a new research approach.
Comments 9: The experiment does not mention any consideration of environmental factors (e.g., soil composition, weather patterns, irrigation methods) in the different regions. These factors could affect the chemical makeup of the mung beans, which might influence the NIR spectra. Their absence may lead to oversights in interpreting the spectral differences purely based on geographical origin.
Response 9: Thanks for the suggestion. As you said, different human and environmental factors in each region have also formed unique regional characteristics. Each region has its own unique soil types and irrigation methods, which are also one of the characteristics of different geographical origins and contribute to the classification and recognition of geographical origins. For example, the irrigation method in Baicheng also constitutes the product characteristics of the geographical origin, so different origins have their own unique soil, weather and other features. Therefore, it ultimately boils down to the fact that agricultural products from different origins have their own near-infrared spectral information. Of course, irrigation and climate in different years may also have a certain impact on the spectrum. In future research, the factor of different years needs to be considered.
Comments 10: Line 141. How were grinded samples stored?
Response 10: Thanks for the suggestion. All mung bean samples were placed in sealed bags and stored in a constant temperature and humidity box (temperature 25 ℃, relative humidity 45%) until the end of the experiment (Line 174-177).
Comments 11: Line 154. How were the “most optimal spectra” selected?
Response 11: Thanks for the suggestion. We revised the sentence (Line 189-190).
Comments 12: Several preprocessing techniques are used (e.g., Standardization, Savitzky-Golay, SNV, MSC), but there is no rationale provided for why these specific methods were chosen or how they improve the results. It's unclear whether other techniques were considered or why the listed methods were deemed optimal for mung beans.
Response 12: Thanks for the suggestion. Through Wu et al. 's comparison of NIR and Raman spectra combined with chemometrics for the classification and quantification of mung beans (Vigna radiata L.) of different origins. It mainly included Standardization (SS), Savitzky-Golay (SG), Standard Normal Variate (SNV), and Multiplicative Scatter Correction (MSC), among others. Standardization centralizes the mean value of the original spectrum and eliminate the influence of deviations caused by dimensional differences, minor self-variations, and significant numerical disparities. Savitzky-Golay of the original spectrum aids in reducing noise, enhancing the uniformity of the spectrum, and thereby mitigating the impact of noise. SNV is primarily employed to diminish interference arising from solid particle mass, surface scattering, and variations in the optical path during the measurement of spectral signal data. MSC is designed to eliminate the effects of scattering phenomena caused by uneven particle sizes or differences in sample particle sizes in the spectral data. Consequently, this study primarily selected these four methods for the spectral preprocessing of near-infrared (NIR) original spectra (Line 195-206).
Comments 13: The experimental section discusses creating discriminant models using PCA and KNN but does not mention how the model performance was evaluated beyond dividing data into training and validation sets. Details about key metrics like accuracy, precision, recall, F1score, or confusion matrices are missing. These metrics are crucial to assessing model robustness and generalizability.
Response 13: Thanks for the suggestion and we added this section: In order to evaluate the discriminative model constructed based on near-infrared spectroscopy data, the trained model is used to predict the data in the test set and obtain the performance indicators of the model (Precision, Recall, Accuracy, F1-score). And explained its performance indicators one by one (Line 337-352).
By analyzing the evaluation indicators of near-infrared discrimination models for mung beans from different origins, the accuracy of the models established for mung bean samples from different origins was 98.25%, which is close to the accuracy of origin discrimination; The F1-score values are all above 0.97 (Table 4).
Comments 14: A basic split into training and validation sets is mentioned, but cross-validation (such as k-fold) could improve the rigor of model evaluation. Not using cross-validation can result in overfitting, especially with smaller datasets.
Response 14: Thanks for the suggestion. We used the K-cross validation method to divide the obtained sample quantity (n=200, 400, 600) dataset into four subsets. One subset (50, 100, 150 respectively) was designated as the validation set to assess the model's discrimination accuracy, whereas the remaining three subsets (150, 300, 450 respectively) were used as the training set (sample set) to ascertain the model's precision and reliability. Similarly, using the same method to divide the obtained sample traceability scope (Tailai-Sishui, Tailai-Baicheng and Tailai-Dorbod Mongol Autonomous) dataset into four subsets. One subset (Tailai-Sishui (75), Tailai-Baicheng (75) and Tailai-Dorbod Mongol Autonomous (75)) was designated as the validation set to assess the model's discrimination accuracy, whereas the remaining three subsets (Tailai-Sishui (225), Tailai-Baicheng (225) and Tailai-Dorbod Mongol Autonomous (225)) were used as the training set (sample set) to ascertain the model's precision and reliability. Similarly, using the same method to divide the obtained sample shelf life (90 d, 180 d, 270 d, 360 d, 90-180-270-360 d) dataset into four subsets. One subset (90 d (150), 180 d (150), 270 d (150), 360 d (150), 90-180-270-360 d (600)) was designated as the validation set to assess the model's discrimination accuracy, whereas the remaining three subsets (90 d (450), 180 d (450), 270 d (450), 360 d (450), 90-180-270-360 d (1800)) were used as the training set (sample set) to ascertain the model's precision and reliability.
Comments 15: The sample sizes for PCA modeling (n=200, 400, 600) are mentioned, but there's no explanation for why these specific numbers were chosen. Was there an analysis to ensure these numbers were statistically significant for the geographic regions studied? Additionally, no randomization or repetition of sampling is mentioned, which could introduce sampling bias.
Response 15: Thanks for the suggestion. In response to the influence of sample quantity on the traceability and discrimination results of mung bean origin, the selected diverse sample quantity information comes from the collection of 600 spectral data using near-infrared spectroscopy technology on raw spectra of mung bean samples from different origins. Based on the references, most studies use a proportional increase (1∶2∶3) in sample quantity to analyze the impact of different origins on traceability discrimination results. Therefore, a random selection of samples (n=200, 400, 600) belonging to the smaller, medium, and larger sample quantities was chosen from the near-infrared spectral data collected from different origins of mung beans to reduce errors (Line 363-369).
In addition, by using principal component analysis to analyze the influence of sample quantity on the traceability discrimination results of mung bean origin, the cumulative variance contribution rate of different sample quantities obtained was subjected to significance analysis, and the results showed significant differences (Table 5). At the same time, based on the results of principal component extraction, it can be seen that the discrimination accuracy of larger sample (n=600) and smaller sample (n=200) has increased by 4.67%, which is also meaningful for the accuracy of near-infrared spectroscopy traceability discrimination.
Comments 16: The model's performance with different sample sizes (200, 400, 600) is only assessed in a specific geographic scope. It would be important to test the model on a more diverse set of regions or agricultural products to determine its broader applicability.
Response 16: Thanks for the suggestion. In this study, mung bean samples with different sample quantities were collected within a specific region. If the range is wider, the experimental results obtained may be different. As you said, the traceability discrimination results obtained from different sample quantities represented by different origins are also different. However, what can be obtained from this study is that the different sample quantities within our specific region affect the accuracy of traceability discrimination. The more samples there are, the more representative they are. Therefore, the stability of the model established has also been improved.
Comments 17: Line 241. The text mentions using the KNN method but does not specify the number of neighbors (k) used in the analysis or any other parameters. Providing this detail would help in understanding how the model's performance was achieved. While K-fold cross-validation is mentioned, it would be useful to know how many folds were used, and if any other validation techniques were considered or used.
Response 17: Thanks for the suggestion. There are not many parameters that need to be adjusted when performing KNN, only the value of k needs to be set. The common methods for determining the value of k include empirical methods and cross validation methods. For the setting of k value, different private values (k=1, 2, 3, …, n) have also been chosen in different studies. This study uses cross validation method. By using K-fold cross validation (k=4), the obtained 600 spectral dataset was divided into four subsets (three subsets (k-1) were selected as the training set and one subset was selected as the validation set). By training with different k values on the training set, the k value with the highest accuracy in evaluating the model on the validation set was found. This method can determine the value of k based on the characteristics of the data itself, which is relatively scientific and reasonable (Line 311-316).
Comments 18: Line 281. There’s mention of PCA results but not a clear connection on how each preprocessing method influenced the PCA results. A more detailed analysis of how each preprocessing method affected the PCA output could provide deeper insights.
Response 18: Thanks for the suggestion. We added the different preprocessing method influenced the PCA results. Standardization finds the principal component information that best represents the feature data in principal component analysis, ensuring its contribution. SG improves the signal-to-noise ratio of spectra, making the characteristic peaks more prominent, which is helpful for near-infrared spectral feature recognition and quantitative analysis. SNV converts data into a standard normal distribution, emphasizing the normality of the data distribution. It is possible to objectively analyze the relationship between near-infrared characteristic spectra, in order to identify the principal component information that truly dominates data changes. MSC scales the data to a new minimum and maximum value, preserve the distribution pattern of the original near-infrared spectral data, improve the predictive ability of the model, and ensure the effectiveness and accuracy of its model. In summary, different spectral preprocessing methods have varying impacts on the results of PCA.
Especially in data analysis and machine learning, preprocessing (such as Standardization, SS, SG, SNV, MSC, etc.) is a key step in data preprocessing, which can significantly affect the extraction of principal component feature data and the establishment of model prediction results (Line 267-276).
Comments 19: Please offer more in-depth statistical analysis and significance testing. The differences in accuracy for various sample sizes are presented, but there is no mention of statistical tests to determine whether these differences are significant. This would be important for evaluating the reliability of the results. Were any statistical tests (e.g., t-tests, ANOVA) performed to determine if the accuracy differences between sample sizes are statistically significant?
Response 19: Thanks for the suggestion. An ANOVA analysis was conducted on the impact of sample quantity on the results of origin tracing discrimination during the preliminary experimental process. The result shows significant differences. In addition, using the accuracy of origin discrimination as an indicator, the overall accuracy of origin discrimination for mung beans with a larger sample quantity (n= 600) was 98.67%, while the overall accuracy of origin discrimination for mung beans with a smaller sample quantity (n= 200) increased by 4.67%. It can be seen that it is meaningful to apply the factor of different sample quantities to the study of traceability in different geographical origins.
Comments 20: Expanding the comparison with other studies. The text briefly mentions studies by Li et al. and Qiu et al. but does not explain how these studies are directly comparable or what lessons can be drawn from their methodologies. Further discussion on differences in methodologies and datasets would add depth to the comparison.
Response 20: Thanks for the suggestion. We expanded the comparison with other studies. Qiu et al. quantified the mineral element content in 40, 80, and 120 mung bean samples using ICP-MS and established the discrimination accuracies of discrimination of mung bean samples of different origins using principal component and PLS discriminant analyses based on the models established using different quantities of samples. The results showed that the accuracy of the mineral element discrimination model established using larger sample quantity (n= 120) for different mung bean origins was 96.70%, while the accuracy of the discrimination model with smaller sample quantity (n= 40) was higher than 1.70%. This article is consistent with the above research results. The accuracy of the near-infrared discrimination model for different mung bean origins established using larger sample quantity (n= 600) is 98.67%, and the accuracy of the discrimination model with smaller sample quantity (n= 200) is higher than 1.97%. This indicates that the higher the sample quantity, the higher the accuracy of the model prediction (Line 419-427).
Comments 21: The text does not mention the computational requirements for processing and modeling with different sample sizes. This is important for other researchers who might want to replicate the study, especially when handling large datasets.
Response 21: In big data statistics, due to the extremely large amount of data, the classification criteria for sample quantity may different significantly from the definition in traditional statistics. For the classification of "large sample", "medium sample" and "small sample": The large sample refers to data points ranging from millions to billions or more; The medium sample consists of datasets points ranging from tens of thousands to millions; The small sample consists of several hundred to tens of thousands of datasets.
In this study, small, medium, and large sample quantities were selected from the "small sample" in statistics to analyze whether different sample quantities have an impact on the effectiveness of origin tracing discrimination. However, there is currently no specific standard for the sample quantity from the origin of agricultural products, so this article is also exploring the impact of sample quantity on origin traceability discrimination. The research results indicate that the sample quantity has an impact on the results of origin traceability discrimination. The more samples there are, the more representative the feature information covering the place of origin is, and the more accurate the discrimination model for place of origin samples is. Therefore, the choice of sample quantity is an important factor affecting the traceability discrimination model for agricultural products.
Comments 22: The findings are specific to mung beans, but there is no discussion about whether the results can be applied to other agricultural products. A broader discussionabout the potential generalizability of the results would enhance the impact of the study.
Response 22: Thanks for the suggestion. We added the disscussion about the potential generalizability of the results. As you said, this study investigated the influence of different geographical origins of the mung bean sample traceability scope, shelf life, and other factors on the model discrimination results, studying the sample traceability scope remains challenging in a limited scope of origin traceability and accurate discrimination. In addition, relevant scholars have also studied the influence of the traceability scope of different origins on their discrimination accuracy. The traceability scope of agricultural products from different origins is different, and their discrimination results are also different. The differences in origin, climate, soil, water and other factors between the traceability scope and local samples also affect the discrimination research of traceability of different agricultural product origins (Line 674-693).
Furthermore, this study demonstrated a correlation between the overall correct discrimination rate of mung bean origins from different regions and a discrimination model based on the shelf life of different samples. In the process of sample discrimination, over time, the components such as moisture and protein in the sample may undergo slow changes, which have an impact on the collection of near-infrared spectroscopy data and the discrimination results of its established model. Even within the same shelf life, storage conditions in different origins (such as humidity, temperature, light, etc.) may have different effects on its discrimination performance (Line 696-704).
Comments 23: References should be revised. References should include abbreviated journal titles.
Response 23: Thanks for the suggestion. We revised the references (abbreviated journal titles).

Reviewer 3 Report
Comments and Suggestions for Authors
The paper entitled “Sample quantity, traceability scope, and shelf life on the discrimination of near-infrared origin traceability of mung beans”, authored by Ming-Ming Chen, Yan Song, Yan-Long Li, Xin-Yue Sun, Feng Zuo and Li-Li Qian, describes the practical application of instrumental and chemometric methods in discrimination of mung beans samples of different origin.
The paper is nicely written and well-organized. While reading the paper, one gets the impression that the authors invested considerable effort in this study. The paper should be accepted for publication after some minor corrections listed below in the comments.
THE TITLE: The title is well thought out and fully reflects the aim of the study.
ABSTRACT: The abstract is well-written. The authors pointed out the main purpose of the study as well as the most significant findings.
INTRODUCTION: Here, the authors cited the most relevant studies and clearly pointed out what has been done so far in the similar fields. In the last paragraph, the authors described the main purpose and the aims of the study.
MATERIALS AND METHODS: In this section, authors clearly described all the materials and methods used in the study. I would just suggest a bit detailed description how PCA was conducted. That was the minimum Eigenvalue for selection of relevant PCs, as well as if the data used in the analysis were normalized.
RESULTS AND DISCUSSION: Generally, the results are thoroughly described and clearly presented. Here, I have some suggestion for the improvement of the results presentation:
- Graphs in Fig. 2 are not so clear (the resolution should be improved). Also, the PC2 axis is not fully visible. I suggest adding the variance % newt to the axes, it is easier to follow.
- Graphs in Fig. 3 should be also improved.
- Fig. 5, 6 and 8 – the variance % should be added next to the axes.
- Why the authors limited the PCA to only 3 PCs?
CONCLUSIONS: Very well written. The authors pointed out the main findings, results and limitations of the study. Also, further investigations are suggested as well.
Author Response
Comments 1: THE TITLE: The title is well thought out and fully reflects the aim of the study.
Response 1: Thanks for your comments and recognition.
Comments 2: ABSTRACT: The abstract is well-written. The authors pointed out the main purpose of the study as well as the most significant findings.
Response 2: Thanks for your affirmation. We did spend a long time researching this article, and the results were quite satisfactory.
Comments 3: INTRODUCTION: Here, the authors cited the most relevant studies and clearly pointed out what has been done so far in the similar fields. In the last paragraph, the authors described the main purpose and the aims of the study.
Response 3: Thanks for your comments and recognition. We have searched for many relevant articles and elaborated on the current research status of our research content.
Comments 4: MATERIALS AND METHODS: In this section, authors clearly described all the materials and methods used in the study. I would just suggest a bit detailed description how PCA was conducted. That was the minimum Eigenvalue for selection of relevant PCs, as well as if the data used in the analysis were normalized.
Response 4: Thanks for the suggestion. We added this section: The commonly used software for principal component analysis is IBM SPSS Statistics 26.0, Origin 2021 Pro, Matlab, Python, etc.. In Python, the PCA class in the Scikit-learn can be used to implement its analysis. Upon applying PCA to the analytical data for four geographical origins, mung bean samples could be preliminarily clustered (the first three PCs), as well as the data used in the analysis were normalized. (Line 246-254).
Also, as described detail in Section 3.1.2, to determine the optimal number of principal components, the principal component variables were screened, and draw the principal component rubble map (Figure 2), the first three principal components can be extracted (Line 292-295).
Comments 5: RESULTS AND DISCUSSION: Generally, the results are thoroughly described and clearly presented. Here, I have some suggestion for the improvement of the results presentation:
(1) Graphs in Fig. 2 are not so clear (the resolution should be improved). Also, the PC2 axis is not fully visible. I suggest adding the variance % newt to the axes, it is easier to follow.
Response 5 (1): Thanks for the suggestion and we have adjusted the resolution of Figure 3, and added the variance % newt to the axes (Figure 3).
(2) Graphs in Fig. 3 should be also improved.
Response 5 (2): Thanks for the suggestion. We have adjusted the resolution of Figure 4.
(3) Fig. 5, 6 and 8–the variance % should be added next to the axes.
Response 5 (3): Thanks for the suggestion and we added the variance % newt to the axes (Figure 6, 7, and 9).
(4) Why the authors limited the PCA to only 3 PCs?
Response 5 (4): To determine the optimal number of principal components, the principal component variables were screened, and draw the principal component rubble map (Figure 2), the first three principal components can be extracted. Its cumulative variance contribution rate is 98.16%>80.00%, which conforms to the basic principle of extracting principal component information. Therefore, the first three principal component variables were selected as the subsequent influencing factors for the study of the effect of near-infrared spectroscopy origin tracing discrimination (Line 292-295).
Comments 6: CONCLUSIONS: Very well written. The authors pointed out the main findings, results and limitations of the study. Also, further investigations are suggested as well.
Response 6: Thanks for your comments and recognition. We did spend a long time researching this article, and the results were quite satisfactory.
